# Quantitative PCR from human genomic DNA: The determination of gene copy numbers for congenital adrenal hyperplasia and RCCX copy number variation

**Márton Doleschall**[1] *, **Ottó Darvasi**[2†], **Zoltán Herold**[3], **Zoltán Doleschall**[4],
**Gábor Nyirő**[1,3], **Anikó Somogyi**[5], **Péter Igaz**[1,3,6], **Attila Patócs**[2,5,7]

**1** Molecular Medicine Research Group, Eotvos Lorand Research Network and Semmelweis University, Budapest, Hungary, **2** Hereditary Tumours Research Group, Eotvos Lorand Research Network and Semmelweis University, Budapest, Hungary, **3** Department of Internal Medicine and Oncology, Faculty of Medicine, Semmelweis University, Budapest, Hungary, **4** Department of Pathogenetics, National Institute of Oncology, Budapest, Hungary, **5** Department of Internal Medicine and Hematology, Faculty of Medicine, Semmelweis University, Budapest, Hungary, **6** Department of Endocrinology, Faculty of Medicine, Semmelweis University, Budapest, Hungary, **7** Department of Molecular Genetics, National Institute of Oncology, Budapest, Hungary

† Deceased.

* marton.doleschall@gmx.com

**Data Availability Statement:** All relevant data are within the manuscript and its Supporting Information files. The minimal data set will be also

## Abstract

Quantitative PCR (qPCR) is used for the determination of gene copy number (GCN). GCNs contribute to human disorders, and characterize copy number variation (CNV). The single laboratory method validations of duplex qPCR assays with hydrolysis probes on *CYP21A1P* and *CYP21A2* genes, residing a CNV (RCCX CNV) and related to congenital adrenal hyperplasia, were performed using 46 human genomic DNA samples. We also performed the verifications on 5 qPCR assays for the genetic elements of RCCX CNV; *C4A*, *C4B*, CNV breakpoint, HERV-K(C4) CNV deletion and insertion alleles. Precision of each qPCR assay was under 1.01 CV%. Accuracy (relative error) ranged from 4.96±4.08% to 9.91±8.93%. Accuracy was not tightly linked to precision, but was significantly correlated with the efficiency of normalization using the *RPPH1* internal reference gene (Spearman's ρ: 0.793–0.940, p>0.0001), ambiguity (ρ = 0.671, p = 0.029) and misclassification (ρ = 0.769, p = 0.009). A strong genomic matrix effect was observed, and target-singleplex (one target gene in one assay) qPCR was able to appropriately differentiate 2 GCN from 3 GCN at best. The analysis of all GCNs from the 7 qPCR assays using a multiplex approach increased the resolution of differentiation, and produced 98% of GCNs unambiguously, and all of which were in 100% concordance with GCNs measured by Southern blot, MLPA and aCGH. We conclude that the use of an internal (in one assay with the target gene) reference gene, the use of allele-specific primers or probes, and the multiplex approach (in one assay or different assays) are crucial for GCN determination using qPCR or other methods.

available in Zenodo database (DOI: 10.5281/
zenodo.6780358) after the publication of the
current paper.

**Funding:** The current research was supported by
Semmelweis Science and Innovation Fund to MD
(STIA-KF-17) and Hungarian Scientific Research
Fund to AP (K125231). MD was supported by
Janos Bolyai Research Scholarship from the
Hungarian Academy of Sciences, and the UNKP-
19-4 New National Excellence Program of the
Ministry for Innovation and Technology. The
funders had no role in study design, data collection
and analysis, decision to publish, or preparation of
the manuscript.

**Competing interests:** The authors have declared
that no competing interests exist.

## Introduction

Quantitative PCR (qPCR) was originally developed for virus quantification [1], and has
recently attracted more attention owing to the SARS-CoV-2 pandemic. It is most often used
for the quantification of mRNA levels [2], but the gene copy number (GCN) determination in
diploid genomes has gained benefit from it for a long time [3]. GCN is the number of repeats
of a gene in one or two sets of chromosomes. The vast majority of the genes occurs twice in a
diploid genome, but the copy numbers of some genes can differ from two. GCN is a non-nega-
tive whole number, and the "integer GCN" term is used when this characteristic is emphasized.
Variations in GCN contributes to both rare genetic disorders and common diseases in humans
[4].

   The qPCR for human GCN determination can be distinguished from qPCR for gene
expression by some key features: 1.) Genomic DNA is the template. The template complexity,
which can reduce the performance of qPCR [5], is much greater in genomic DNA than in total
mRNA of a particular tissue: The haploid human genome consists of 3.1 billion base pairs, and
millions of base pairs differ between two random haploid chromosome sets [6], while a few
hundred genes account for 50% of transcripts in most human tissues [7] covering only a cou-
ple of hundred thousand base pairs in total length.

   2.) Limit of detection (LOD) is not crucial. The absolute copy number of a target gene in a
DNA sample is proportional to the absolute number of haploid chromosome sets, which can
be approximately calculated from the mass of genomic DNA in the sample. The absolute num-
ber of haploid chromosome sets can be more accurately determined by the quantitative mea-
surement of a reference gene, which invariably occurs once in each haploid chromosome set.
The ratio of absolute copy numbers of a target gene and a reference gene in the DNA sample
of a subject is identical to the ratio of the copies of target and reference genes in two haploid
chromosome sets of a diploid cell. The ratios of the target and reference genes is not condi-
tional on the amount of genomic DNA in a sample, and the GCN of a target gene is easily cal-
culated from this ratio since the GCN of a reference gene is always two in a diploid cell.
Therefore, the amount of genomic DNA in a measurement also does not influence GCN (in
theory), and can be chosen to be conveniently above the limit of detection (LOD).

   3.) The differentiation between greater consecutive GCNs is difficult. The quantification
cycle ($C_q$) is determined by qPCR to characterize the absolute copy number of a gene in reality.
$C_q$ is proportional to a relatively short DNA sequence specific to a target or a reference gene,
and GCN is calculated from $C_q$s related to the target and reference genes. GCN determined by
qPCR can be called "measured GCN", and can be a positive real number (for example, a ratio-
nal number), not necessarily a non-negative whole number. The relationship between $C_q$ and
GCN can be described by the equation: $C_q$(target gene)-$C_q$(reference gene) = -(($\log_2$(GCN)/
$\log_2$(2))-1. The reference gene $C_q$ is constant in theory, and therefore only the target gene $C_q$
determine GCN. The theoretical difference between two target gene $C_q$s derived from two con-
secutive GCNs will approach zero if GCN approaches infinity. This means that the theoretical
difference of two target gene $C_q$s is $\infty$ between 0 and 1 GCN, $\Delta C_q = 1$ between 2 and 1 GCNs,
$\Delta C_q = 0.585$ between 3 and 2 GCNs, $\Delta C_q = 0.415$ between 4 and 3 GCNs, $\Delta C_q = 0.322$ between
5 and 4 GCNs, and so on. Therefore, it becomes more and more difficult to differentiate the
greater consecutive GCN, which presents the key problem of qPCR as well as other molecular
biology methods for GCN determination.

   4.) The inaccurately measured GCNs can be easily identified in the majority of cases. Ambi-
guity is the state of a measured GCN which is not close enough to an integer GCN to assign
unequivocally the measured GCN to the integer GCN. The measured GCN is a continuous
variable, and therefore a measured GCN can be about halfway between two integer GCNs,

which clearly indicates the inaccuracy of the particular measurement. The distribution of several measured GCNs derived from the same integer GCN approaches a normal distribution, resulting in the majority of the measured GCNs around the real integer GCN (unambiguous GCNs), some measured GCNs between the real GCN and an adjacent integer GCN (ambiguous GCNs) and a few measured GCNs around the adjacent integer GCNs (misclassified GCNs).

Returning to the key problem, there are two techniques allowing GCN methods to extend beyond this limitation: 1.) The use of allele-specific primers or probes. Paralogous genes or gene variants are the copies of the same gene (or very similar ones), which have high DNA sequence similarity, and are located at the different loci of a haploid chromosome set. There are very often sequence differences between the paralogous gene variants which can be targeted in an allele-specific way, and the total GCN of the gene variants can be decomposed into smaller GCNs. 2.) The use of several probes (in one or different assays) for the same target gene. The integer GCN of a target gene can be estimated from the measured GCNs of more DNA sequences that are parts of the target gene, which can make the estimation more reliable. This estimation of the integer GCN can be based on the personal decision of an operator [8], a simple mathematical measure such as arithmetic mean [9] or a more complex statistical method such as a classifier.

The above-mentioned features and techniques are well illustrated by RCCX copy number variation (CNV, S1 Table). RCCX CNV usually consists of 1–3 tandem repeats of a DNA segment on one chromosome, and each DNA segment harbors 2 complete genes, complement component 4 (*C4*) and steroid 21-hydroxylase (*CYP21*) [10, 11]. Therefore, the copy numbers of the RCCX CNV segment, *C4* and *CYP21* are identical, and no exception to this has been described yet. Both genes have 2–2 paralogous gene variants. *C4* genes are sorted into *C4A* and *C4B* genes differing in 5 nucleotides in exon 26. The *CYP21* genes sort into a functioning gene (*CYP21A2*) and a pseudogene (*CYP21A1P*) having several sequence differences. *CYP21A2* contributes to the steroidogenesis of adrenal glands, and its mutations cause congenital adrenal hyperplasia (CAH) [12]. An additional sequence difference of *C4* genes derived from a virus insertion in intron 9 (called human endogenous retrovirus K (HERV-K (C4) CNV)), and a RCCX CNV breakpoint, where two CNV segments are joined, are currently being researched [13]. Multiplex ligation-dependent probe amplification (MLPA) for the GCNs of *CYP21* genes is commercially available, and is recognized as an appropriate methods in the genetic testing of CAH [14]. MLPA uses multiple *CYP21A1P*- and *CYP21A2*-specific probes, and special statistical methods followed by the final evaluation of integer GCNs by the operator.

The GCNs in RCCX CNV are also determined by qPCR based on allele-specific primers or probes [13, 15–17]. However, these qPCR assays are singleplex for target genes (target-singleplex) in stark contrast to MLPA, and none of their published documentations completely meets the requirements of "minimum information for publication of quantitative real-time PCR experiments" (MIQE) [18, 19]. In this study we therefore aimed to simultaneously assess the performance of 7 qPCR assays for the GCN determination of the genetic elements of RCCX CNV according to MIQE. Verifications were performed on *C4A*, *C4B*, HERV-K(C4) CNV deletion, HERV-K(C4) CNV insertion, and RCCX CNV breakpoint qPCR assays in the current study because some information has been published on these assay performances (S2 Table), whereas single laboratory method validations [20] were completed on *CYP21* qPCR assays. Furthermore, our goals were to examine the optimal laboratory strategy of qPCR for GCN determination in general, and the fit-for-purpose of qPCR assay for *CYP21A2* GCN determination in the genetic testing of CAH.

## Materials and methods

### DNA samples

The qPCR validation and verification processes were completely in accordance with MIQE (S1 File). The current research was conducted with the approval by the National Scientific and Ethical Committee, Medical Research Council of Hungary (TUKEB, ETT), approval number 4457/2012/EKU. Written informed consent was given by all of the study subjects. Genomic DNA samples were extracted from whole blood of 10 healthy subjects, 11 patients with CAH and 5 patients with non-functioning adrenal incidentaloma (NFAI) using a Qiagen QIAcube instrument (S3 Table) with Qiagen QIAamp DNA blood mini kit or a Roche DNA isolation kit for mammalian blood (S1 File). Genomic DNA samples were also purchased from the International Histocompatibility Working Group (IHWG). Purchased DNA samples derive from 18 healthy subjects of the HapMap European reference population (CEU) [21] and 2 HLA homozygous cell lines, COX and QBL. Purchased DNA samples were isolated by IHWG using 5-Prime ArchivePure DNA cell/tissue kit, and their nominal concentrations were 100 ng/µl. DNA samples of COX and QBL were applied in a 1:1 mixture. An SD039 reference DNA sample for MLPA with a nominal concentration of 10 ng/µl, which is included in MRC Holland SALSA MLPA probe mix P050 CAH, was also used. The concentration and purity of all DNA stock solutions were determined by a Thermo Fisher Scientific (TFS) NanoDrop 2000 spectrophotometer and a TFS Qubit 1 fluorometer with Qubit dsDNA high sensitivity assay kit, and DNA integrity was checked on 0.7 m/V% agarose gels.

### Positive controls, samples for calibration curves and study groups

DNA working solutions with 5 ng/µl DNA concentration were separately diluted from the stock solutions of the DNA samples for 3 replicate measurements, except for ones for positive controls (more than 3 separately diluted working solutions) and for the calibration curve (a series of dilutions). DNA samples derived from our own subjects were divided based on DNA quality into "good quality" (A260/A280>1.8 and A260/A230>2.0 and no sign of DNA degradation) and "bad quality" study groups (n = 10). The SD039 reference sample for MLPA was assigned to the "good quality" group (n = 17). The DNA samples purchased from IHWG were labeled as the "population" group (n = 19).

### Measurements

The applied custom primers (Integrated DNA Technologies) and hydrolysis probes (produced by TFS) for *C4* genes [16], *CYP21* genes [15], HERV-K(C4) CNV alleles and RCCX CNV breakpoint [13] (S4 Table) have been previously published. All these hydrolysis probes had a 5'-fluorochrome, 6-carboxyfluorescein (FAM) reporter, and a 3'-nonfluorescent quencher and a 3'-minor groove binder. The mix for qPCR measurements (S5 Table) also contained ribonuclease P RNA component H1 (*RPPH1*) internal reference gene included in TFS TaqMan human RNase P copy number reference assay with a probe labeled with 2′-chloro-7′-phenyl-1,4-dichloro-6-carboxy-fluorescein (VIC), and TFS TaqMan fast advanced master mix (FAMM). The concentrations of custom primers and probes were slightly different in different qPCR assays (S5 Table). The same qPCR profile according to the manual of FAMM was used for all 7 qPCR assays. The qPCR measurements were carried on a TFS QuantStudio 7 qPCR instrument (QS7) with TFS QuantStudio software v1.2 except for some experiments with SYBR Green and robustness experiments (see more later). The three DNA working solutions of a DNA sample were separately measured for a target gene on different days except for ones for positive controls. We measured 3 replicates from the same and 3 replicates from the

different DNA working solutions of a DNA sample for positive control on each of the three days. The quantification threshold of the *RPPH1* reference gene was always 0.1 for relative quantification. The quantification thresholds for the replicate measurements of each target gene were tuned in a way that the average of relative errors (REs) of measurements (based on a preliminary calculation) in a replicate measurement of "good quality" and "population" study group equaled approximately zero (S6 Table). Therefore, all measurements of these study groups (n = 36) were taken into account, instead of the use of arbitrarily selected reference samples and $\Delta\Delta C_t$ method. There was no need for the further normalization or correction of the $C_q$ values, and there was no $C_q$ optimization for precision, calibration curve or the variances of RE.

The melting curve analyses were performed with Bioline SensiFAST SYBR Lo-ROX kit on GS7. The micro-capillary electrophoreses of duplex qPCR reactions were carried out by Agilent Bioanalyzer 2100 instrument with Agilent Bioanalyzer high sensitivity DNA kit. MRC Holland SALSA MLPA EK1 reagent kit and probe mix P050 CAH kit on TFS ProFlex PCR and TFS 3130 capillary electrophoresis instruments were used for the measurements of MLPA, and MRC Holland Coffalyser software v140721 for the calculation of MLPA results. Robustness experiments were performed using a Roche LightCycler 1.0 instrument with Roche Light-Cycler FastStart DNA master SYBR green I reagent, QS7 with TFS TaqMan universal master mix II without uracil-N-glycosylase (UMM2) or a TFS 7500 Fast qPCR instrument (7500F) with FAMM. All qPCR reagents in robustness experiments were used according to the manuals of the manufacturers.

## Calculation and statistics

Non-specific PCR products were *in silico* predicted by Primer Blast [22], and the secondary structures of PCR products were *in silico* determined by UNAFold [23]. The limit of detection was estimated according to Hubaux and Vos [24], and statistical metrics were used based on MIQE (S6 Table). Statistical analyses were performed with R v4.0.2 [25] and SPSS v26. Normal distribution was tested by Shapiro–Wilk (SW) test. Fisher's exact test (FE), Student's t-test, Wilcoxon test, ANOVA with Tukey post-hoc test, Kruskal-Wallis (KW) test with Dunn post-hoc test, Pearson's correlation, Spearman's rank correlation and Levene's test were used for basic statistics. Tests were two-tailed, p-values were corrected with the false discovery rate (FDR) method, and $p < 0.05$ was considered as statistically significant. A linear mixed-effect model of the R package lme4 [26] was applied to the analyses of slopes of calibration curves. Integer GCNs were estimated from measured GCNs by a machine learning classifier, the linear discriminant analysis (LDA).

## Results

### Parameters of genomic DNA samples

DNA concentration, A260/A280, A260/A230 were determined, and DNA integrity was also tested. The genomic DNA samples purchased from IHWG were assigned to the "population" study group (n = 19). The DNA samples of the subjects, enrolled by us, were divided based on DNA quality into "good quality" (A260/A280≥1.8 and A260/A230≥2.0 and no sign of DNA degradation) and "bad quality" study groups (n = 10). A reference sample (SD039) was also assigned to the "good quality" group (n = 17). The mean and SD belonging to the A260/A280 and A260/A230 quality parameters of the stock solutions of genomic DNA sample were 1.848 ±0.043 and 2.064±0.483 in the „good quality", 1.817±0.049 and 1.899±0.182 in the „population", and 1.787±0.046 and 2.152±0.089 in the „bad quality" study group (S1 File). The "population" group included 5 DNA samples of 19 which had slightly lower A260/A280 values than

1.8. However, all samples were sample intact in the "population" and "good quality" groups. The DNA samples in the "bad quality" study group were partially degraded. The concentrations of DNA stock solutions measured by Qubit were significantly lower (Wilcoxon test: p<0.0001) than those by Nanodrop (S1 Fig) agreeing with a previous findings [27]. Nevertheless, genomic DNA concentrations theoretically affect neither precision nor accuracy in qPCR as long as DNA concentration is in the linear range of the measurement.

### Analytical specificity

Putative non-specific PCR products were found only at the primer pair of HERV-K(C4) CNV insertion target element using Primer-BLAST (S7 Table). Non-specific PCR product was observed only at the primer pairs of *C4A* and *C4B* target genes by melting curve analyses (S2 Fig), and the micro-capillary electrophoresis of duplex qPCR reactions confirmed this finding. Nevertheless, the same PCR product is amplified from both *C4A* and *C4B* gene variants, and allele-specific hydrolysis probes have also discriminated between the different target sequences of the mixed PCR products generated by the same primer pair in a previous study [28]. A couple of nucleotide differences in a target sequence can completely block the binding of the non-specific probe, and a non-specific PCR product can bind a specific probe by chance with very low probability. Therefore, the non-specific PCR product does not necessarily distort the quantitative PCR performance of *C4* assays.

### Matrix effect of genomic DNA

The $C_q$ values (S1 File) of the *RPPH1* reference gene from different assays could be compared, because the concentration of *RPPH1* reagents, qPCR running parameters and the quantification threshold for *RPPH1* were identical in all assays. There were no significant differences between the *RPPH1* $C_q$s in replicate measurements (SW FDR: p = 0.676–0.768; ANOVA: p = 0.632; Tukey: p = 0.717–1.000) (S3 Fig), but significant differences (SW FDR: p = 0.002–0.958, 3 significant ones out of 46; KW: p<0.0001; Dunn: 39.7% significant pairs) were observed between DNA samples (Fig 1). The matrix effect of genomic DNA (sample-to-sample variation) could cause the differing $C_q$ means of different samples, because the causes, derived from the quality and quantity of DNA solutions, could be ruled out; the DNA quality was completely checked, the DNA extractions were performed in 2 independent laboratories using different methods, the concentrations of DNA stock solutions were double-checked with 2 different methods, and $C_q$s were measured from 3 independent diluted working solution series.

### Linearity, PCR efficiency and analytical sensitivity

Separate calibration curves were measured using 3 different genomic DNA samples in each assay, and 5 DNA samples were selected in total 1.) to ensure the GCNs of the target genes were as diverse as possible, and 2.) to measure each sample at least 3 times (S4 and S5 Figs). The effects of samples and different assays on the regression slopes were examined by a linear mixed-effects model (S8 Table). Samples had twice as large an effect on the standard deviation (SD) of slopes than the assays. The effect of samples on slope deviations is more likely to arise from the matrix effect of genomic DNA (template complexity) than from the deviated PCR inhibition of different samples because there is no difference in the deviation of $C_q$s from the lines of calibration curves between low and high DNA concentrations.

　　The average PCR efficiencies in different assays were around 1 (S9 Table), and there were no significant differences between them (SW FDR: p = 0.857, ANOVA: p = 0.333, Tukey: p = 0.356–1.000 for target genes, SW FDR: p = 0.687–0.983, ANOVA: p = 0.604, Tukey: p = 0.627–1.000 for reference genes). The PCR efficiencies of target and *RPPH1* pairs from the

## *RPPH1* reference gene by DNA sample

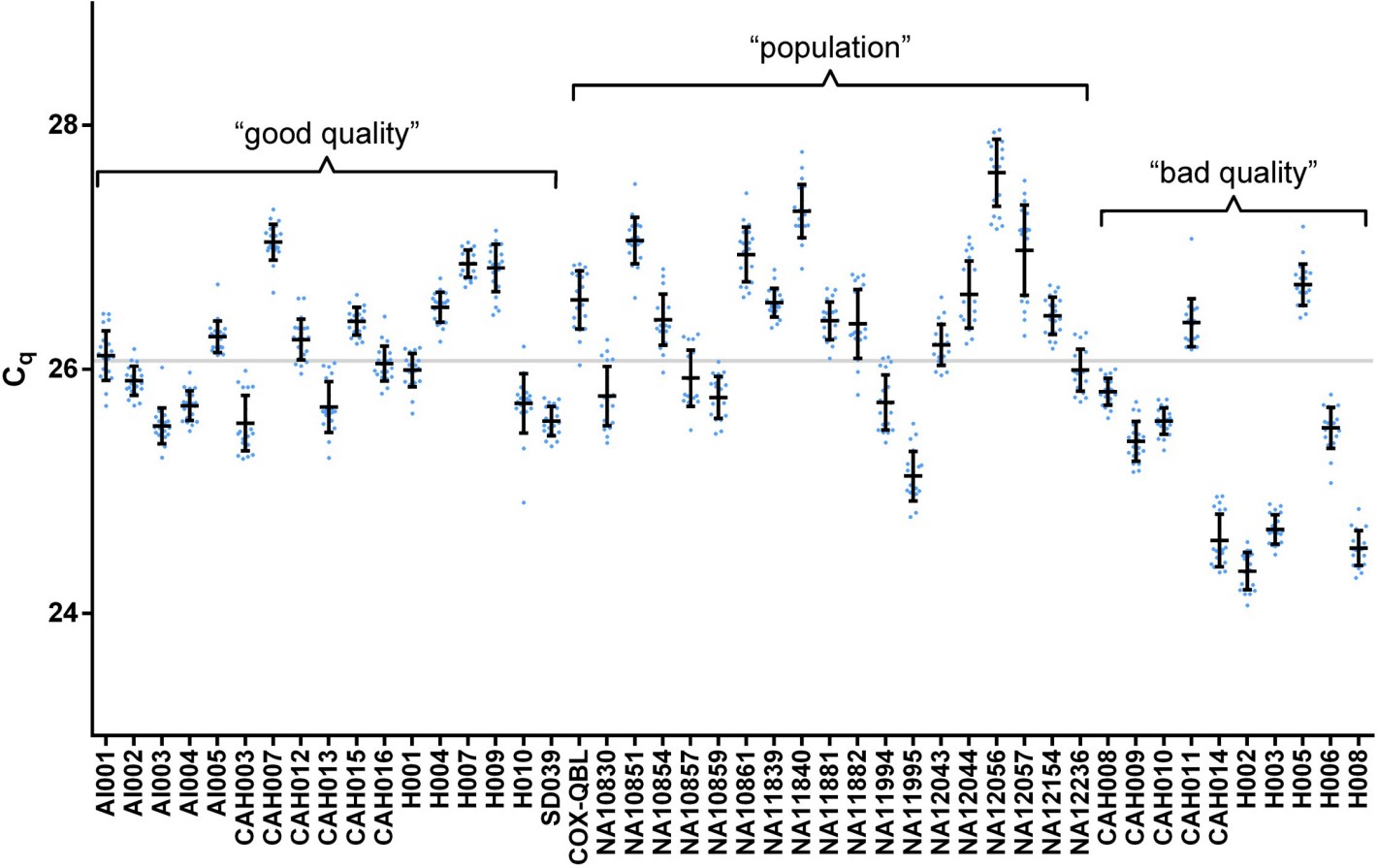

**Fig 1. $C_q$ values (N = 966) of the *RPPH1* reference gene grouped by DNA samples.** The measurements of only one, predefined DNA working solution were taken into account for a calibration curve or a positive control DNA sample. A light blue dot indicates one *RPPH1* Cq value. Means and standard deviations are indicated by bars. The mean of *RPPH1* $C_q$ (26.0724) is indicated by a horizontal grey line.

same assays and DNA samples showed a strong and significant correlation (Spearman's ρ = 0.7426, p = 0.0001) (S6 Fig), indicating that *RPPH1* effectively compensated for the matrix effect on PCR efficiency. Estimated LODs were around the theoretical limit, which seems over-estimated (S9 Table). Nevertheless, the lowest dilution of calibration curves contained approximately 400–1200 copies of genomic template depending on the GCN, and all 63 measurements on this dilution produced adequate $C_q$s and GCNs, indicating that the lowest applied concentration was well above LOD, in agreement with a previous study [29].

### Precision

Repeatability and reproducibility were assessed from the same and separate dilutions of positive control samples, and both of them showed low and quite homogenous coefficient of variation % (CV%) values throughout the assays (S10 Table). Reproducibility values were also assessed in "good quality", "population" and "bad quality" study groups producing comparable results, and the highest pooled CV% was 1.01.

## Gene copy numbers and their concordance

The measured GCNs between ±0.3 of an integer GCN were considered as unambiguous. The ambiguity of GCNs is usually defined by a customarily applied fixed limit, which increases the ambiguous GCNs at higher integer GCNs. For instance, a 10% difference between the integer and measured GCNs means an unambiguous result at a GCN of 2 (1.8 or 2.2 measured GCNs), but an ambiguous one at a GCN of 4 (3.6 or 4.4 measured GCNs) using a ±0.3 fixed limit. The majority of the average measured GCNs of samples in "good quality" and "population" study groups (Fig 2, S1 File) were unambiguous except for the GCNs of the HERV-K (C4) CNV insertion assay (S11 Table). Measured GCNs in the "bad quality" group were markedly ambiguous in *CYP21A1P* and *CYP21A2* assays, implying a high sensitivity of these assays to DNA quality.

There is no reference method or certified reference material for GCN determinations in RCCX CNV. However, MLPA is most often used for *CYP21* genes in the genetic testing of CAH [14]. Furthermore, the GCNs of *C4A* and *C4B* in the CEU human reference population [30, 31] and the full RCCX CNV DNA sequences [32, 33] and GCNs [13] of COX and QBL HLA homozygous cell lines have been determined (S1 File). The GCNs were fully concordant in SD039 and COX-QBL samples, and *C4A*, *C4B* and RCCX CNV breakpoint GCNs were reasonably concordant with the previous results of Southern blot and array comparative genome hybridization (CGH) (S12 Table). *CYP21A1P* and *CYP21A2* GCNs were also suitably concordant with GCNs determined by MLPA (S7 Fig). The precisions of MLPA probes ranged between 12.60–38.62 CV% (S13 Table). The ambiguity was significantly higher for the *CYP21A2* MLPA probe, which recognizes the same insertion allele of a 8 bp genetic variant than *CYP21A2* qPCR assay, compared to the ambiguity of qPCR (35% vs 11% ambiguous GCNs, FE FDR: p = 0.024) in spite of appropriate quality controls of MLPA (S8 Fig). The reproducibility of MLPA was also assessed with the same dilutions of positive control samples in the same way as performed in qPCR assays; The reproducibility of GCNs based on positive controls for β-defensin loci are 6.25 CV% for qPCR and 2.88 CV% for MLPA in a previous study [34], whereas they were 5.08 CV% for *CYP21A1P* qPCR, 3.04 CV% for *CYP21A2* qPCR, 4.84 CV% for *CYP21A1P* MLPA and 7.52 CV% for *CYP21A2* MLPA in the current study.

## Expected gene copy numbers, estimated integer gene copy numbers and consistency

The larger deviation (above 0.4) of a particular measured GCN from the expected GCN calculated by the multiple linear regression of other GCNs in the same sample highlighted the inconsistent results (S9 Fig, S1 File). Integer GCNs were estimated from the measured GCNs by LDA. Total *C4*, total *CYP21*, total HERV-K(C4) CNV GCNs in addition to RCCX CNV breakpoint GCN plus 2 were all equal, and used for the estimation of total integer GCNs. The integer GCN of a paralogous gene was identical to total GCN, where the GCN of the other paralogous gene was 0. The integer GCNs of a paralogous gene pair having both GCNs larger than 0 were estimated from the measured GCNs of the paralogous gene pair and RCCX CNV breakpoint (Fig 3).

The estimations on integer GCNs were unambiguous in 3 cases (1.94%) based on the cross-validation and probabilities of LDA, while the majority of estimations (N = 150, 98.06%) passed the cross-validation (S1 File). All unambiguously estimated integer GCNs were in 100% of concordance with the integer GCNs measured by Southern blot, MLPA and array CGH (N = 137), suggesting that the probabilities and cross-validation of LDA accurately indicated the ambiguity at estimated integer GCNs. Furthermore, all unambiguously estimated integer GCNs showed 100% consistency with each other.

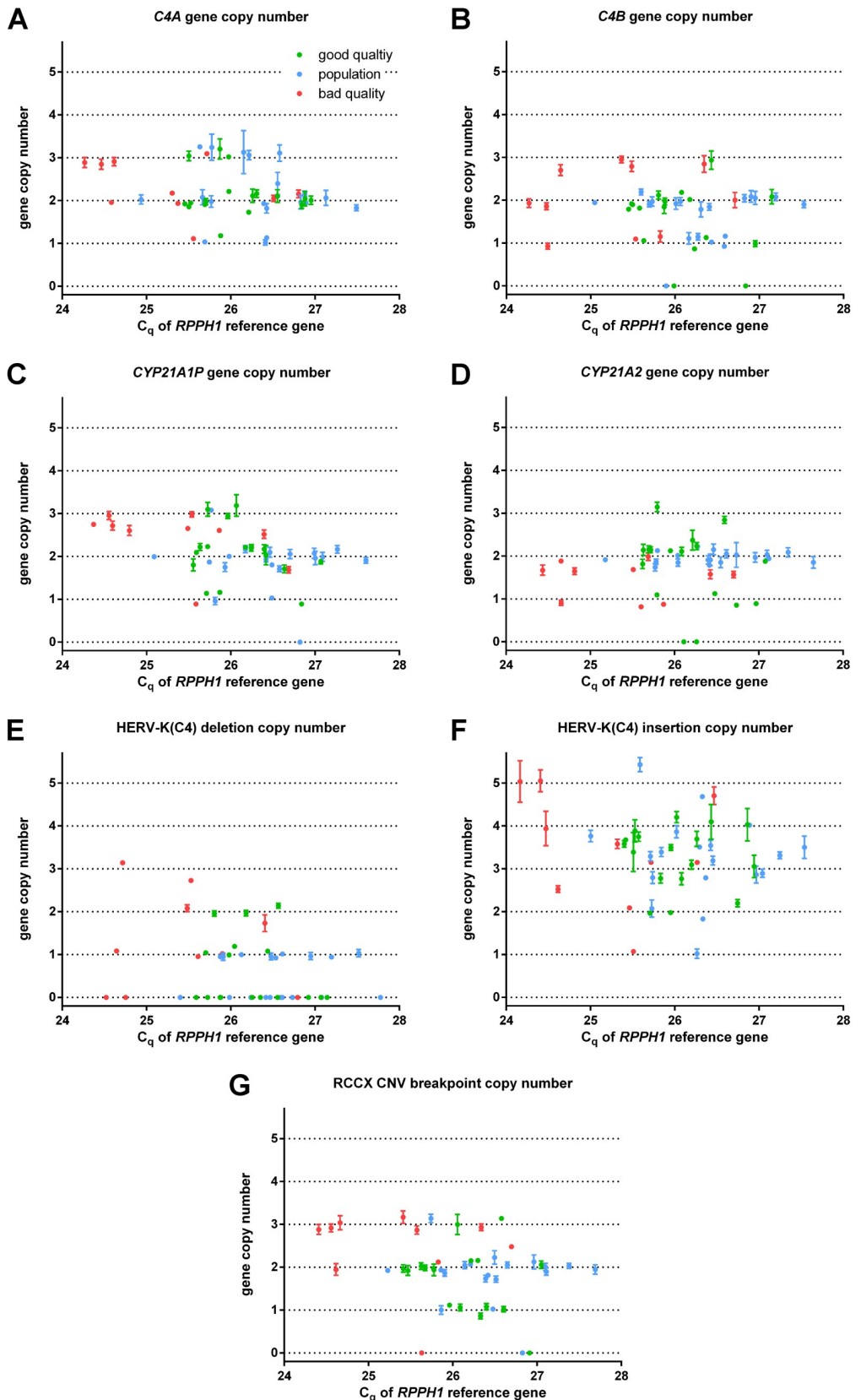

**Fig 2. Average measured gene copy numbers (GCNs) of samples in different qPCR assays for RCCX CNV.** Bars indicate standard deviation.

## Estimated accuracy

The RE of measurements in study groups sometimes significantly deviated from a normal distribution and from each other (Fig 4). However, there was no such significant difference (SW FDR: 0.055–0.892; ANOVA: 0.552–0.972) in REs grouped by replicate measurements

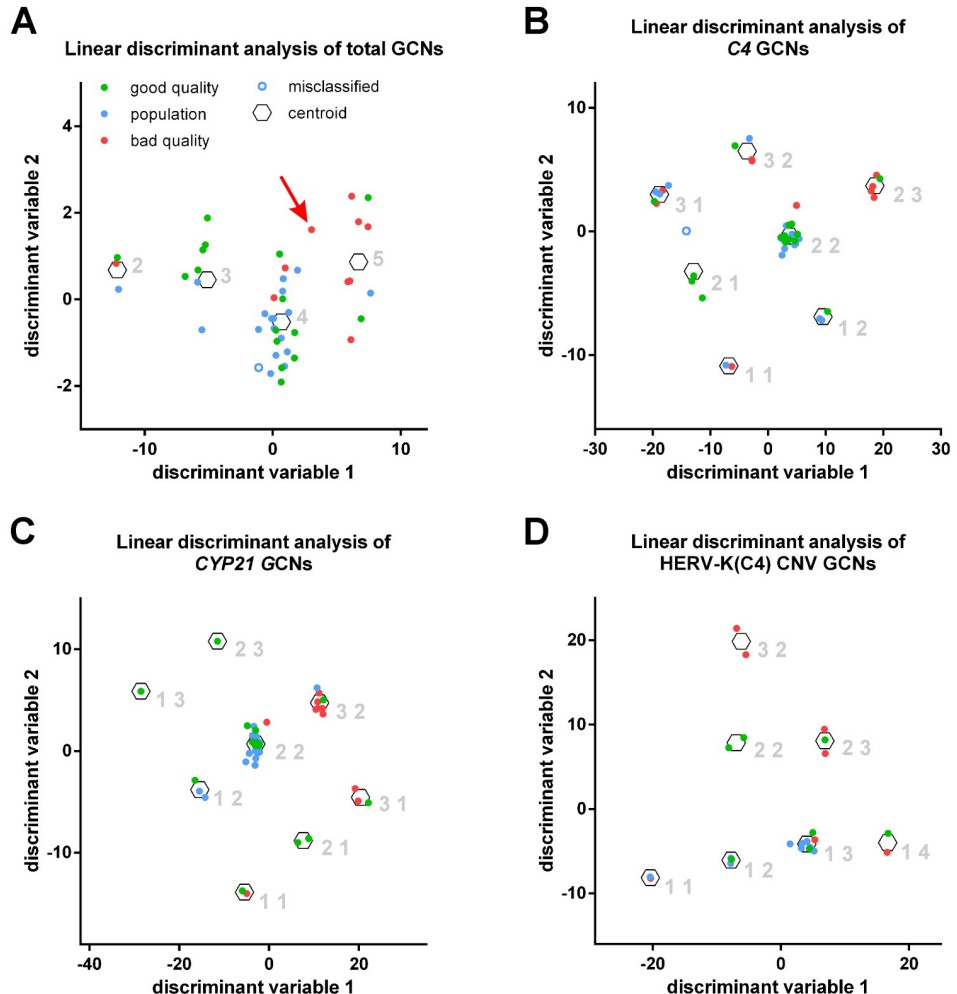

**Fig 3. Locations of samples based on discriminant scores generated by linear discriminant analyses (LDA). A.**) LDA of total gene copy numbers (GCNs). The GCNs of *C4* genes, *CYP21* genes, the alleles of HERV-K(C4) CNV and the copy number of RCCX CNV breakpoint + 2 equal each other as well as the copy number of RCCX CNV segment due to genome biology reasons. Grey numbers indicate the total integer GCNs of the particular clusters. **B.**) LDA of *C4* genes. First grey number indicates the integer *C4A* GCNs of the particular clusters, and the second one indicates the integer *C4B* GCNs. **C.**) LDA of *CYP21* genes. First grey number indicates the integer *CYP21A1P* GCNs of the particular clusters, and the second one indicates the integer *CYP21A2* GCNs. **D.**) LDA of the alleles of HERV-K(C4) CNV. First grey number indicates the integer HERV-K(C4) CNV deletion GCNs of the particular clusters, and the second one indicates the integer HERV-K(C4) CNV insertion GCNs. Blue empty circle indicates the NA11839 sample, which was correctly classified for total GCN in spite of its incorrect input class, but its cross-validation of LDA showed a different class for *C4* genes. Red arrow indicates the H005 sample, which also received a different class of total GCN by cross-validation than its original class.

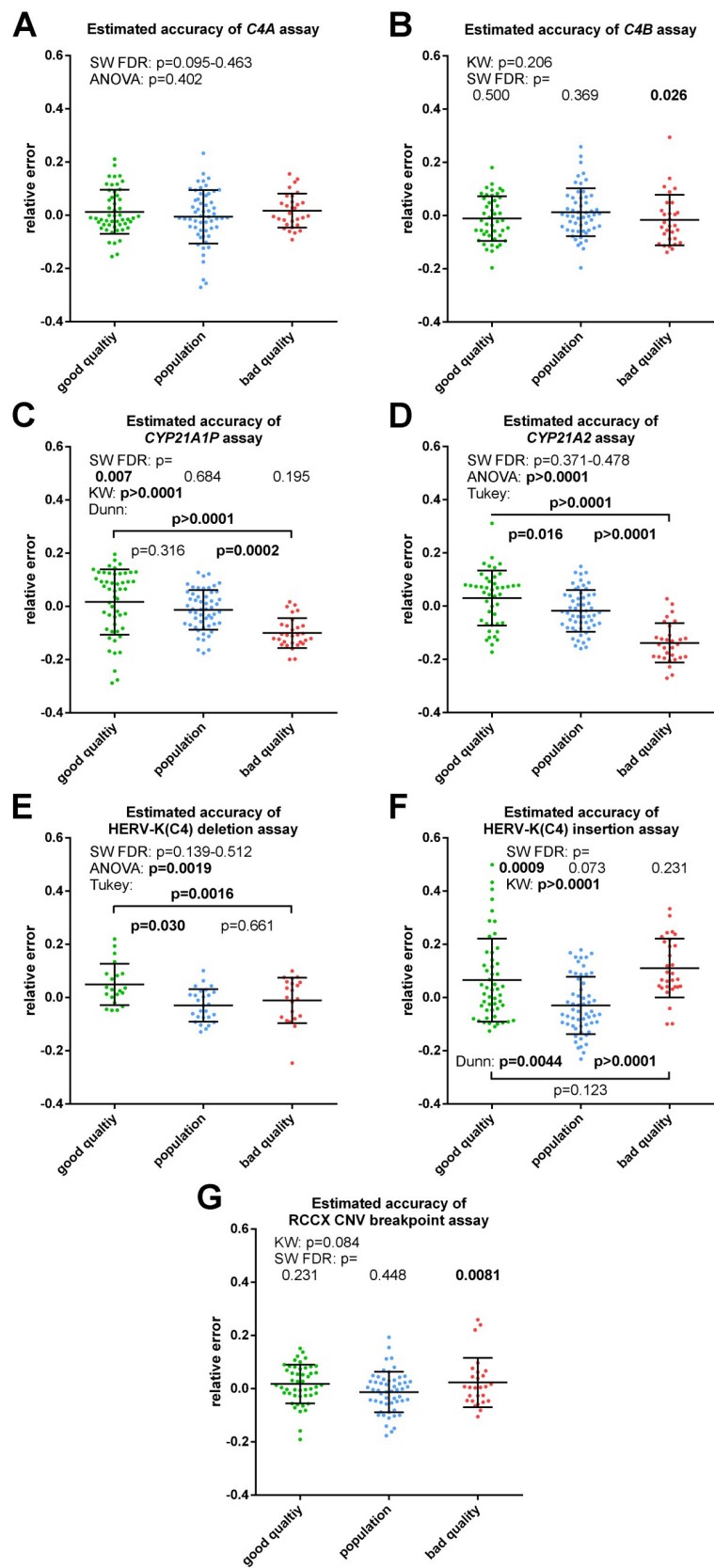

**Fig 4. Estimated accuracy of different qPCR assays for RCCX CNV grouped by study group.** Estimated accuracy is expressed by the relative error of the qPCR measurements. Samples with rarer GCNs were selected for "good quality" and "bad quality" groups, which may cause the difference between these groups and the "population" group in HERV-K(C4) CNV assays, whereas the lower GCNs of the "bad quality" study group in *CYP21A1P* and *CYP21A2* assays was already observed at measured GCNs. Relative errors were not calculated for the samples with 0 GCNs in the particular assay. Bars indicate means and standard deviation. SW—Shapiro–Wilk test, FDR—false discovery rate method for multiple testing correction, KW—Kruskal-Wallis test.

(S10 Fig). Significant differences were between REs in the same assay grouped by GCNs (S11 Fig), although a clear tendency could not be observed. The REs grouped by DNA samples (Fig 5) reflected a significant matrix effect of genomic DNA (SW FDR: p = 0.060–0.987; ANOVA: p<0.0001; Tukey: 21.5% significant pairs), but the matrix effect on accuracy seemed to show a lesser extent than that on *RPPH1* C$_q$s.

The means and SDs belonging to the absolute values of average REs of samples in the assays were between 4.96±4.08% and 9.91±8.93% (S12 Fig). The distributions of average REs fitted to normal distributions. The normal distribution of a particular assay, characterized by mean and

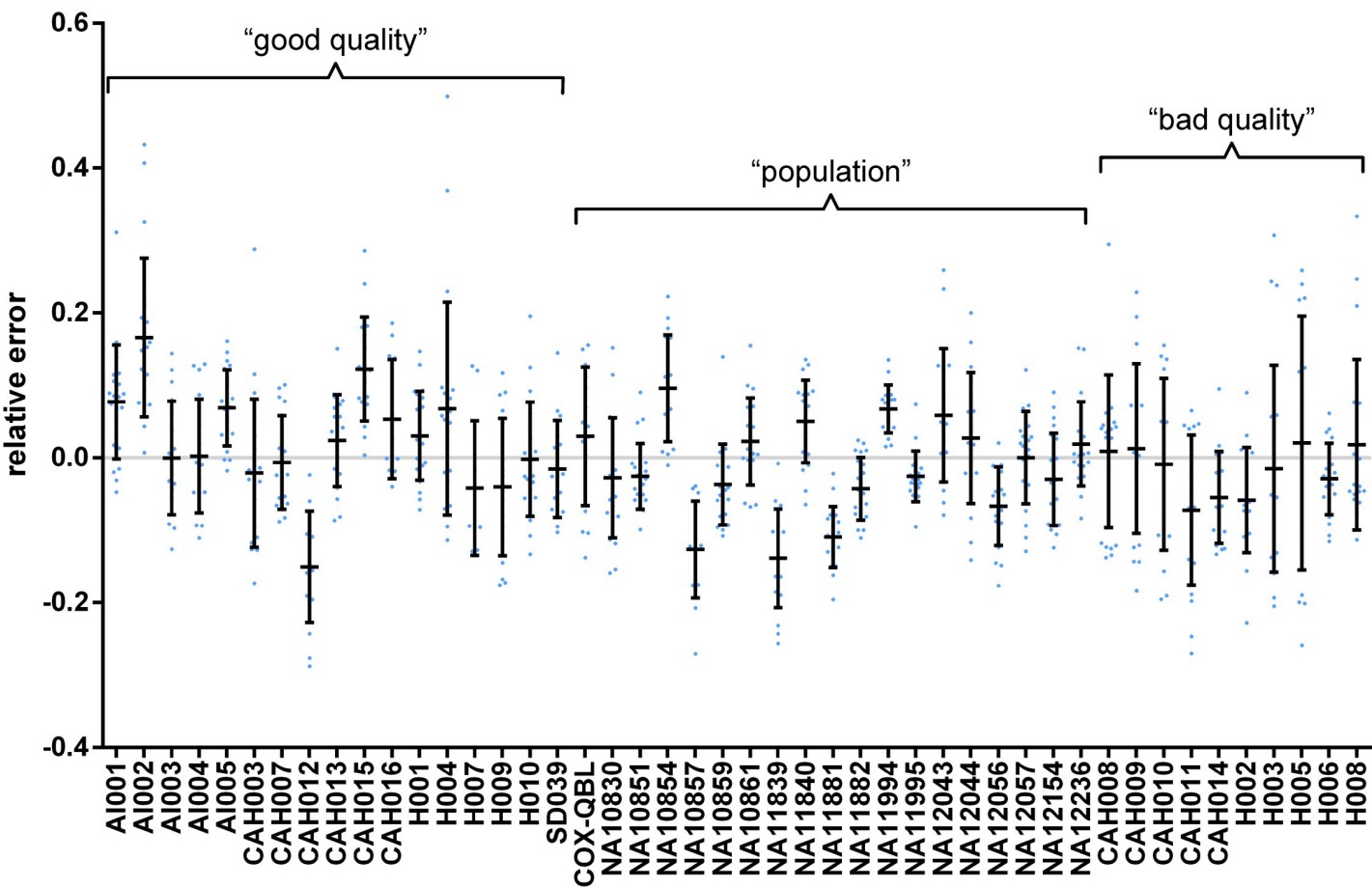

**Fig 5. Estimated accuracy grouped by samples.** Estimated accuracy is expressed by the relative error of the qPCR measurements. Relative errors were not calculated for the samples with 0 GCNs. Bars indicate means and standard deviation.

**Table 1. Estimated ambiguity and misclassification rates of different qPCR assays for RCCX CNV at different gene copy numbers (GCNs).**

| | C4A assay | C4B assay | CYP21A1P assay | CYP21A2 assay | HERV-K(C4) CNV deletion assay | HERV-K(C4) CNV insertion assay | RCCX CNV breakpoint assay |
|---|---|---|---|---|---|---|---|
| ambiguity at 1 GCN | 0.02% | 0.01% | 0.15% | 0.03% | >0.01% | 2.57% | >0.01% |
| ambiguity at 2 GCN | 6.16% | 4.94% | 11.25% | 7.07% | 2.78% | 26.47% | 2.05% |
| ambiguity at 3 GCN | 21.27% | 19.01% | 29.00% | 22.83% | 14.24% | 45.71% | 12.23% |
| ambiguity at 4 GCN | 35.00% | 32.58% | 42.74% | 36.6% | 27.12% | 57.71% | 24.65% |
| misclassification at 1 GCN | >0.01% | >0.01% | >0.01% | >0.01% | >0.01% | >0.01% | >0.01% |
| misclassification at 2 GCN | >0.01% | >0.01% | 0.02% | >0.01% | >0.01% | 0.93% | >0.01% |
| misclassification at 3 GCN | 0.36% | 0.23% | 1.37% | 0.49% | 0.06% | 8.28% | 0.03% |
| misclassification at 4 GCN | 2.92% | 2.19% | 6.41% | 3.50% | 1.02% | 19.32% | 0.68% |

Estimations were performed based on the normal distributions (characterized by mean and standard deviation) of average relative errors of samples.

SD, was assumed around every integer GCNs of the particular assay to estimate the ambiguity and misclassification rates. The estimated ambiguity and misclassification rates of assays (Table 1) were in accordance with observed ambiguity and concordance (S11 and S12 Tables). The estimated ambiguities were fairly moderate at a GCN of 2 in the assays with better performance, whereas the estimated misclassifications were sufficiently low at a GCN of 3. The ambiguity and misclassification rates were also estimated based on the normal distributions calculated by the average REs of each GCN in a particular assay (S14 Table).

### Robustness of *CYP21A1P* and *CYP21A2* assays

The robustness was screened through precision in *CYP21* genes, which was reasonably similar to the general setup except when using different qPCR reagents and instruments (S15 Table). Performance was assessed with UMM2 and a 7500F (S13–S16 Figs and S16–S20 Tables). The estimated ambiguity and misclassification rates with UMM2 were similar to the better ones of the assays measured by the general setup, despite the higher imprecision and worse linearity. The PCR efficiencies between *CYP21* target and *RPPH1* showed a higher correlation (S17 Fig) with UMM2 than with 7500F. Normalized root-mean-square error (NRMSE) is a common measure of the differences between a predicted value and a measured one. NRMSE was used to characterize how efficiently the $C_q$s of a target gene follows the $C_q$s of the *RPPH1*. The reproducibility values of target and *RPPH1* were significantly correlated from sample to sample in all *CYP21* assays, as well as NRMSE and accuracy, but there was no correlation between precision and accuracy (S21 Table). The same relationships could be observed in other assays (S22 Table). Moreover, the SDs of accuracies (the SDs of the average REs of samples) in the assays from the default and robustness experiments of "good quality" and "population" study groups were significantly correlated with observed ambiguities and misclassifications (Spearman's $\rho = 0.671$, $p = 0.029$ and $\rho = 0.769$, $p = 0.009$).

### Discussion

Despite the observation that the performance parameters of 7 different qPCR assays in the current study vary, the number of assays was large enough and the conditions of measurements

were homogenous enough to draw generalized conclusions. The precision based on $C_q$s and calibration curve parameters were not strongly linked to the accuracy of GCNs. The HERV-K (C4) CNV insertion assay showed similarly good precision and calibration curve parameters to other assays, but its accuracy (RE) had a high SD. In contrast, worse precision and calibration curves were observed in the assays of *CYP21* genes measured with UMM2, but their accuracies were similar to the assays with high performance. The accuracy is directly associated with ambiguity and misclassification, and therefore accuracy should be considered as the key performance parameter of qPCR for GCN. The matrix effect of genomic DNA observed at *RPPH1* $C_q$s, calibration curves, and even accuracy to a lesser extent, seemed to have a great impact on the performance of qPCR for GCN. The higher effectiveness of the normalization to the *RPPH1* reference gene was correlated with the higher accuracy, and the normalization probably contributed to the reduced matrix effect on accuracy. If ambiguity of around 5% and misclassification under 1% are considered acceptable, the methodological limit of a singleplex qPCR assay for a target gene with 3 replicates proved to be a GCN of 2 for ambiguity and a GCN of 3 for misclassification in general. A lower methodological limit of GCN [35], and a similar one [36] have been described in the literature of qPCR. At any rate, the higher range of integer GCNs occurring in patients or a study population often exceeds the methodological GCN limit of target-singleplex qPCR, producing enough ambiguous GCNs and misclassifications that we should question the fit-for-purpose at higher GCNs.

The usage of multiple target sequences from a multiplex assay [37] or separate assays [38] can overcome the GCN limitation of the target-singleplex method. The measured GCN of the multiple target genetic elements were classified using LDA as a multiplex approach. The majority of integer GCNs (98.06%) estimated from measured GCNs passed the cross-validation of LDA and were qualified as unambiguous. Furthermore, all these unambiguously estimated integer GCNs were in 100% concordance with the integer GCN estimations from MLPA and the findings of Southern blot and array CGH from previous studies [30, 31]. LDA was highly effective, even using the ambiguous data of qPCR assays with lower accuracy, such as HERV-K(C4) CNV insertion assay and DNA samples with bad quality. The low sample size of a GCN class limits the performance of the classifier, so it may be worth using a reference set enriched with samples having rarer GCNs. The multiplex approach could render qPCR for the genetic elements of RCCX CNV very effective, however, it would be interesting to see how this multiplex approach for qPCR tackles a target region with higher average GCN than that found in RCCX CNV.

Several studies [29, 34, 39–41] contrast one molecular biology method for GCN with another, often having the ambition to pronounce one of them more advanced or suitable. However, the final conclusions of these studies are controversial, and it is difficult to draw a general conclusion because: 1.) The performances are compared using only a few metrics. GCNs and their concordance between the examined methods are the typical levels of comparison. Methodological differences can hamper the comparison at multiple levels; for instance, the measurement from a single replicate does not allow for the use of many standard performance metrics such as CV%. 2.) Performance metrics are poorly assessed or documented. This is well-illustrated by qPCR experiments where usually only a fraction of performance metrics required by MIQE are published. Ambiguity between measured and integer GCNs is seldom stated explicitly, but it can be assessed in the majority of GCN determination methods, and the performance of a method can be inferred from it. 3.) Study conclusions are drawn from one or a couple of assays. The performance of a GCN method is limited from the direction of high performance due to theoretical or practical reasons, but it can be mediocre to any extent owing to the poor design or execution. Therefore, an assay of a particular method with higher performance characterizes better the particular method than one with lower

performance. For instance, the reproducibility of GCN in MLPA for CAH in the current study was a little bit worse than the reproducibility in MLPA for β-defensin in a previous article [34], and the MLPA results for β-defensin more aptly characterize MLPA in general. 4.) The performance of a GCN method is only compared to other methods, not to the fitness for purpose. The target-singleplex performance of MLPA for the same genetic variant also detected by *CYP21A2* qPCR assay were lower, but an inappropriate fit-for-purpose does not ensue from this. The real strength of MLPA is the multiplex approach, which provides appropriate final integer GCN results, and the relatively simple procedure. In addition, the MLPA for the genetic test of CAH has some potential to identify CAH mutations and chimeric *CYP21* gene variants in the same assay [42].

Southern blot was the first GCN determination method for RCCX CNV [43], and uses a multiplex approach because the unlabeled genomic DNA fragment pattern bound to the membrane can be examined with several probes for the elements of RCCX CNV in succession. The disadvantages of Southern blot include high labor intensity, high time demand, and only semi-quantitative GCN results based on a human operator's evaluation [44], decreasing its suitability for the genetic test of CAH. Array CGH is a high-throughput method, but its high labor intensity, high time demand and high cost do not fit to the needs of CAH laboratories, where the vast majority of array CGH results would not be used. Other GCN determination methods based on the multiple elements of RCCX CNV such as the paralog ratio test and the high resolution melting PCR have been described [30, 45]. Digital PCR has been used for the GCN determination of *C4* paralogous gene variants and HERV-K(C4) CNV [46], but the methodological performance of the digital PCR has not been evaluated on the genetic elements of RCCX CNV yet.

The *CYP21A2* qPCR assay in the current study produces a reasonable number of ambiguous results at a GCN of 2, and the measurements of ambiguous GCNs can be conveniently repeated because the labor intensity and time demand of qPCR is low. Lower GCNs (0 and 1) are frequently examined for genetic testing, and misclassification for these GCNs is very low. A GCN of 3 seldom has to be examined, and therefore, the high ambiguity at a GCN of 3 is acceptable, as is the low level of misclassification. Furthermore, the analysis of CAH mutations has to be performed in the genetic testing of CAH, and this should correspond to GCNs, indicating the possible misclassification at a GCN of 3 [47]. Overall, therefore, we suggest that, the target-singleplex *CYP21A2* qPCR assay fits for the purpose of the genetic testing of CAH.

## Supporting information

**S1 Fig. The relationship between the concentrations of DNA stock solutions measured by NanoDrop 2000 spectrophotometer and Qubit 1 fluorometer.** Pearson's correlation coefficient is indicated by "r". The data of one predefined dilution (first of separately diluted ones) of each sample for positive control and calibration curve was included in the study groups. Repeatability values of DNA concentration determinations based on positive control samples were 3.93 CV% for NanoDrop and 2.58 CV% for Qubit, and reproducibility values were 3.99 CV% and 4.22 CV%, respectively. The correlation of all samples between NanoDrop and Qubit was high (Pearson's r = 0.887, p<0.0001), although the concentrations of DNA stock solutions measured by Qubit were significantly lower (Wilcoxon test: p<0.0001). Pearson correlation coefficients (r) between NanoDrop and Qubit values in the "bad quality" study group suggest that DNA quality influenced DNA concentration measurements.
(TIF)

**S2 Fig. Melting curve analyses of quantitative PCR primers for different target genes using positive control samples.** Rn is the normalized fluorescence of the reporter dye. Red arrow

shows the peak of a non-specific product.
(TIF)

**S3 Fig. C$_q$ values (N = 966) of the *RPPH1* reference gene grouped by target genes and replicate measurements.** The measurements of only one, predetermined DNA working solution were taken into account for a calibration curve or a positive control DNA sample. A light blue dot indicates one *RPPH1* C$_q$ value. Means and standard deviations are indicated by bars. The mean of *RPPH1* C$_q$ (26.0724) is indicated by a horizontal grey line. Replicate measurement 1, 2 and 3 are indicated by p1, p2 and p3.
(TIF)

**S4 Fig. Calibration curves of target genetic elements.** N$_{temp}$ is the genomic copy number of the particular target genetic elements, which is conditional on the amount of genomic DNA in the series of dilutions (2.5, 5, 10, 20, 40 and 80 ng total DNA in a measurement) and the gene copy number of target elements in the diploid genome (1–4). CI means 95% confidence interval. The CIs of the lines are not depicted because they would be too close the lines to discern.
(TIF)

**S5 Fig. Calibration curves of the *RPPH1* reference gene from different qPCR assays.** N$_{temp}$ is the genomic copy number of the target genetic elements of the *RPPH1* gene, which is conditional on the amount of genomic DNA in the series of dilutions (2.5, 5, 10, 20, 40 and 80 ng total DNA in a measurement). CI means 95% confidence interval. The CIs of the lines are not depicted because they would be too close the lines to discern.
(TIF)

**S6 Fig. Relationship between the PCR efficiencies of target genes and corresponding *RPPH1* reference genes of each sample for calibration curves.** Black line is a simple linear regression. Dotted line indicates the 95% confidence intervals.
(TIF)

**S7 Fig. Relationships between gene copy numbers (GCNs) of qPCR assays for *CYP21* genes and MLPA for CAH.** Ambiguity thresholds (±0.3 of integer GCNs for qPCR and according to the manual for MLPA) are indicated by grey dotted lines. Bars indicate the standard deviation of GCNs in the case of qPCR, and the standard deviation of the dosage quotients (equivalent of GCN in MLPA) of different probes for the same *CYP21* gene in the case of MLPA.
(TIF)

**S8 Fig. Relationship between gene copy numbers (GCNs) of qPCR assays and MLPA probes for the same two alleles of an 8 bp deletion variant in *CYP21* genes (rs387906510, c.332_339delGAGACTAC).** Two MLPA hybridization probes (15221-L20261 and 15221-L20262) detect the two alleles of the same 8 bp deletion variant in *CYP21* genes, which is also detected by *CYP21A1P* and *CYP21A2* qPCR assays in the current study. Therefore, these probes and assays are suitable for direct comparison after normalization with an internal reference probe. The MLPA reference probe 16316-L21434 was used for the normalization of the *CYP21A2* probe (15221-L20261) and *CYP21A1P* probe (15221-L20262), because this reference probe showed the highest correlation with the *CYP21* probes and the other MLPA reference probes. Only the GCNs of the replicates of the first replicate measurement were used for qPCR, because MLPA according to the official manual applies to one measurement of each DNA sample by default. The ratios of *CYP21* and reference probes of MLPA were tuned to approximately zero average relative error, as was done for qPCR. Ambiguity thresholds (±0.3 of integer GCNs) are indicated by grey dotted lines.
(TIF)

**S9 Fig. Relationship of measured and expected GCNs in qPCR assays for RCCX CNV.** A multiple regression model was built from all measured GCNs of study groups based on the genomic relations. The expected total GCNs and RCCX CNV breakpoint GCN plus 2 were calculated in each replicate measurement based on the model. Then the expected average GCN of a particular target gene was calculated using the average of corresponding expected total GCNs in proportion to measured GCNs of the particular target genes and its allelic counterpart.
(TIF)

**S10 Fig. Estimated accuracy of different qPCR assays for RCCX CNV grouped by replicate measurements.** Estimated accuracy is expressed as the relative error of the qPCR measurements. Relative errors were not calculated for the samples with 0 GCN in the particular assay. Bars indicate means and standard deviation.
(TIF)

**S11 Fig. Estimated accuracy of different qPCR assays for RCCX CNV grouped by gene copy numbers (GCNs).** Estimated accuracy is expressed as the relative error of the qPCR measurements. Relative errors were not calculated for the samples with 0 GCN in the particular assay. Bars indicate means and standard deviation. All statistical tests were calculated based on "good quality" and "population" study groups. SW—Shapiro–Wilk test, FDR—false discovery rate method for multiple testing correction, KW—Kruskal-Wallis test.
(TIF)

**S12 Fig. Average estimated accuracy of "good quality" and "population" study groups in different qPCR assays for RCCX CNV.** Estimated accuracy is expressed as the average relative error of the samples. The means and SDs of absolute values of average relative errors of the samples are indicated under the p-values of the Shapiro-Wilk test (SW). Relative errors were not calculated for the samples with 0 GCNs in the particular assay. Bars indicate means and standard deviation. FDR—false discovery rate method for multiple testing correction. The variances of average relative errors of samples in were significantly different (Levene's test: p = 0.0026) between assays. However, only the difference between HERV-K(C4) CNV insertion and RCCX CNV breakpoint assays was significant (Levene's test FDR: p = 0.04996) after multiple testing correction.
(TIF)

**S13 Fig. Calibration curves of *CYP21A1P* and *CYP21A2* target and *RPPH1* reference genes for robustness.** $N_{temp}$ is the copy number of the particular target genetic element in a measurement, which is conditional on the amount of genomic DNA in the series of dilutions (2.5, 5, 10, 20, 40 and 80 ng total DNA in a measurement) and the copy number of the particular genetic element in the diploid genome. CI means 95% confidence interval. UMM2—TaqMan universal master mix II, 7500F - 7500 Fast qPCR instrument.
(TIF)

**S14 Fig. Average measured gene copy numbers of samples in *CYP21A1P* and *CYP21A2* qPCR assays for robustness.** Bars indicate standard deviation. UMM2—TaqMan universal master mix II, 7500F - 7500 Fast qPCR instrument.
(TIF)

**S15 Fig. Estimated accuracy of measurements in *CYP21A1P* and *CYP21A2* qPCR assays for robustness grouped by study group.** Estimated accuracy is expressed by the relative error of the qPCR measurements. Relative errors were not calculated for the samples with 0 GCN in

the particular assay. Bars indicate means and standard deviation. UMM2—TaqMan universal master mix II, 7500F - 7500 Fast qPCR instrument.
(TIF)

**S16 Fig. Average estimated accuracy of samples in *CYP21A1P* and *CYP21A2* qPCR assays for robustness.** Estimated accuracy is expressed by the average relative error of the samples. Relative errors were not calculated for the samples with 0 GCN in the particular assay. Bars indicate means and standard deviation. SW—Shapiro–Wilk test, FDR—false discovery rate method for multiple testing correction. UMM2—TaqMan universal master mix II, 7500F - 7500 Fast qPCR instrument.
(TIF)

**S17 Fig. Relationship between the PCR efficiencies of target genes and corresponding *RPPH1* reference genes of each sample in *CYP21A1P* and *CYP21A2* qPCR assays for robustness.** Correlations were made under the assumption that the PCR efficiencies of *CYP21A1P* and *CYP21A2* assays behave in a similar way. Black line is a simple linear regression. Dotted line indicates the 95% confidence intervals. UMM2—TaqMan universal master mix II,7500F - 7500 Fast qPCR instrument.
(TIF)

**S1 Table. Glossary of important terms used in the current article.** There is no official consensus on the terms of ambiguity, concordance, misclassification and consistency, but they have been widely used in the literature of gene copy number determination.
(PDF)

**S2 Table. Performance information of quantitative PCR (qPCR) for gene copy number (GCN) determination in selected literature.** The term "accuracy" is used as a difference between a measured value and a "true" value determined by a reference material or method. "Ambiguity" is a state of a measured GCN not being close enough to an integer GCN to assign it clearly. "Misclassification" is a state of an integer GCN estimated from measured GCN not being identical to the genuine integer GCN. PMID–PubMed ID, avr–average, SD—standard deviation, $\Delta C_q$–$C_q$(target gene)-$C_q$(reference gene), CV%—coefficient of variation %. [1]There are criteria, but no detailed information. [2]It is implicitly stated on an graph. [3]It is expressed in relation to the results of other method(s) for GCN determination.
(PDF)

**S3 Table. The address of the manufacturers of reagents and instruments.**
(PDF)

**S4 Table. Primers and hydrolysis probes used in the current study.** All primers and probes were purified by HPLC. At least one of the primers of a primer pair is bound to an intronic sequence. Allele-specific sites are indicated on the sequences by underscore.
(PDF)

**S5 Table. The concentrations of quantitative PCR reagents in different set-ups.** The RNaseP copy number (CN) reference assay contains the *RPPH1* internal reference gene. A reaction usually contained 10 ng genomic DNA, but the reactions for calibration curves also contained 2.5, 5, 20, 40 or 80 ng genomic DNA. TaqMan fast advanced master mix was used as qPCR reagent with AmpliTaq™ Fast DNA Polymerase. Additional $Mg^{2+}$, additional dNTP and other additives were not added to the reactions. MicroAmp fast 96-well reaction plates (cat. no.: 4346907) were used for qPCR measurements.
(PDF)

**S6 Table. Statistical metrics of the current study.** m(gDNA)–mass of genomic DNA, $M_{nucl\ avr}$–average molar mass of nucleotide, $N_{nucl\ in\ hap\ gen}$–the amount of nucleotide pairs in the haploid human genome, $N_A$–Avogadro's number, CV–coefficient of variation, GCN–gene copy number, $mGCN_{rep}$—measured gene copy number of replicate, $C_q$–quantification cycle, $RE_{mGCN}$–relative error of measured GCN of replicate, eiGCN–estimated integer gene copy number, NRMSE–normalized root-mean-square error.
(PDF)

**S7 Table. Parameters of analytical specificity assessed by *in silico* analysis, melting curve analysis and Agilent Bioanalyzer 2100 (micro-capillary electrophoresis).** Non-specific PCR product was observed only at the primer pairs of *C4A* and *C4B* target genes.
(PDF)

**S8 Table. Analyses of the slopes of calibration curves using a linear mixed-effect model.** Fixed effects were calculated based on all calibration curves of target genes or the *RPPH1* gene, and a random effect of assays or samples were separately modeled. Both fixed effects of the target and *RPPH1* reference genes were calculated from all assays, which can be interpreted as the estimated average slopes. This were very close to a perfect value of -3.322, corresponding to a PCR efficiency of 1. Individual random effects are expressed as a standard deviation from the fixed effect.
(PDF)

**S9 Table. The PCR efficiencies and estimated limit of detection (LOD) of target genetic elements and *RPPH1* reference genes of different qPCR assays in singleplex or duplex PCR reaction.** The PCR efficiencies and estimated LOD were also assessed in the singleplex PCR reaction of each target or reference gene using the AI001 DNA sample. There were no significant differences between average PCR efficiencies from multiplex reactions (SW FDR: p = 0.857, ANOVA: p = 0.333, Tukey: p = 0.356–1.000 for target genes, SW FDR: p = 0.687–0.983, ANOVA: p = 0.604, Tukey: p = 0.627–1.000 for reference genes). LOD was estimated by the Hubaux-Vos method. SD—standard deviation, CI—confidence interval.
(PDF)

**S10 Table. Precisions of target and *RPPH1* reference genes in different qPCR assays for RCCX CNV.** All precisions are calculated by pooled coefficient of variation (CV) and expressed as CV%. Repeatability and reproducibility with same and different dilutions were assessed in positive control samples. The measurements of replicates for reproducibility were performed on different days. Reproducibility in "good quality", "population" and "bad quality" study groups was assessed in samples with GCNs higher than zero. Some tendencies might be observed; the repeatability values from the measurement of the same dilutions tended to be lower than those from measurement of different dilutions, and repeatability values tended to be lower than reproducibility values.
(PDF)

**S11 Table. Ambiguities in different qPCR assays for RCCX CNV.** The measured GCNs between ±0.3 of an integer GCN were considered as unambiguous.
(PDF)

**S12 Table. Concordance in different qPCR assays for RCCX CNV.** Concordance was assessed in the samples with unambiguous gene copy numbers (GCNs) compared to the GCNs from MLPA, Southern blot and array CGH or to the estimated integer GCNs. Percentages indicate the rate of correctly determined GCNs. Percentage is not calculated when n<9.
(PDF)

**S13 Table. Precisions of MLPA probes for CAH.** The peak heights of MLPA probes were determined by Coffalyser software with the default setting. All precisions are calculated by pooled coefficient of variation (CV) and expressed ~~by~~as CV%. The precisions (repeatability and reproducibility) were assessed with the same dilutions of positive control samples in the same way as performed in qPCR assays. Peak heights equaling zero were excluded from calculations.
(PDF)

**S14 Table. Estimated ambiguity and misclassification rates of different qPCR assays for RCCX CNV at different gene copy numbers (GCNs).** Estimations were done based on the means and variances of average relative errors of different GCNs. Estimation were not performed for GCNs with less than 5 samples.
(PDF)

**S15 Table. Precisions of target and *RPPH1* reference genes in *CYP21A1P* and *CYP21A2* qPCR assays for robustness.** The primers of *CYP21A1P* and *CYP21A2* genes were tested in a LightCycler 1.0 instrument with Sybr Green dye, and several qPCR parameters such as total volume, annealing temperature, primer concentration, probe concentration, qPCR reagent (UMM2) and qPCR instrument (7500F) were changed in the FAMM-GS7 system usually used for the current study. All precisions are calculated by pooled coefficient of variation (CV) and expressed as CV%. Repeatability and reproducibility were assessed in positive control samples from the same dilution. FAMM—TaqMan fast advance master mix, UMM2—TaqMan universal master mix II, GS7—GeneStudio 7 qPCR instrument, 7500F - 7500 Fast qPCR instrument.
(PDF)

**S16 Table. The PCR efficiencies of *CYP21A1P* and *CYP21A2* target and *RPPH1* reference genes for robustness.** SD—standard deviation, CI—confidence interval. UMM2—TaqMan universal master mix II, 7500F - 7500 Fast qPCR instrument.
(PDF)

**S17 Table. Ambiguities in *CYP21A1P* and *CYP21A2* qPCR assays for robustness.** UMM2—TaqMan universal master mix II, 7500F - 7500 Fast qPCR instrument.
(PDF)

**S18 Table. Misclassifications in *CYP21A1P* and *CYP21A2* qPCR assays for robustness.** Misclassifiaction was assessed in the samples with unambiguous GCNs compared to estimated integer GCNs. Percentage is not calculated for n<9. UMM2—TaqMan universal master mix II, 7500F - 7500 Fast qPCR instrument.
(PDF)

**S19 Table. Statistical tests for the variances of average relative errors of samples in *CYP21A1P* and *CYP21A2* qPCR assays for robustness.** The variances of average relative errors of samples in the "good quality" and "population" study groups of the assays with UMM2 and 7500F were significantly different. Top left cell of the table contains the test results for all four groups, other cells contains the results between pairs and after multiple testing correction by the false discovery rate method. UMM2—TaqMan universal master mix II, 7500F - 7500 Fast qPCR instrument.
(PDF)

**S20 Table. Estimated ambiguity and misclassification rates of *CYP21A1P* and *CYP21A2* qPCR assays for robustness at different gene copy numbers (GCNs).** Estimations were

made based on the means and standard deviation of average relative errors. UMM2—TaqMan universal master mix II, 7500F - 7500 Fast qPCR instrument.
(PDF)

**S21 Table. Correlation between reproducibility values (coefficients of variance (CVs)) of target $C_q$s of samples, reproducibility values of the reference $C_q$s of samples, normalized root-mean-square errors (NRMSEs) of the target $C_q$s of samples, and accuracy (the average relative errors of samples) in "good quality" and "population" study groups of *CYP21A1P* and *CYP21A2* qPCR assays for robustness and RCCX CNV.** Pooled CV of target and reference gene (reproducibility or inter-assay precision), pooled NRMSD of target Cqs characterizing the normalization of $C_q$s of target genes by $C_q$s of the reference gene, and the standard deviation of the average relative error used for the estimation of ambiguity and misclassification rates are given for information purposes. Correlation was assessed by Spearman's correlation. UMM2—TaqMan universal master mix II, 7500F - 7500 Fast qPCR instrument.
(PDF)

**S22 Table. Correlation between reproducibility values (coefficients of variance (CVs)) of target $C_q$s of samples, reproducibility values of the reference $C_q$s of samples, normalized root-mean-square errors (NRMSEs) of the target $C_q$s of samples and accuracy (the average relative errors of samples) in "good quality" and "population" study groups of *C4A*, *C4B*, HERV-K(C4) CNV deletion, HERV-K(C4) CNV insertion and RCCX CNV breakpoint qPCR assays.** Pooled CV of target and reference gene (reproducibility or interassay precision), pooled NRMSD of target Cqs characterizing the normalization of $C_q$s of target genes by $C_q$s of reference gene, and the standard deviation of average relative error used for the estimation of ambiguity and misclassification rates are given for information purposes. Correlation was assessed by Spearman's correlation.
(PDF)

**S1 File. Sheet 1: MIQE checklist.** All essential information (E) and Desirable information (D) are submitted in the current manuscript. **Sheet 2: Data of DNA stock solutions.** h—healthy, nfai—non-functioning adrenal incidentaloma, sv—simple virilizing congenital adrenal hyperplasia, sw—salt wasting congenital adrenal hyperplasia; g—good quality, p—population, b—bad quality; Q—Qiagen QIAcube wtih QIAamp DNA blood mini kit, R—Roche DNA isolation kit for mammalian blood, 5P - 5 Prime ArchivePure DNA cell/tissue kit; i—intact, sd—slightly degraded, pd—partially degraded. **Sheet 3: Raw Cq values of the main qPCR experiments.** Outliers were not identified using quartile ± 1.5 * interquartile range criterion. NTC—no template control. **Sheet 4: Raw peak heights of MLPA probes.** Coffalyser software with the default settings was used for the determination. **Sheet 5: Raw dosage quotient of *CYP21A1P* and *CYP21A2* MLPA probes.** Coffalyser software with the default settings was used for the determination. **Sheet 6: Raw Cq values for the repeatability and reproducibility values of the robustness experiments.** Outliers were not identified using quartile ± 1.5 * interquartile range criterion. NTC—no template control; FAMM—TaqMan fast advance master mix, UMM2—TaqMan universal master mix II, GS7—GeneStudio 7 qPCR instrument, 7500F - 7500 Fast qPCR instrument. **Sheet 7: Raw Cq values for the detailed *CYP21* robustness experiments.** Outliers were not identified using quartile ± 1.5 * interquartile range criterion. NTC—no template control; FAMM—TaqMan fast advance master mix, UMM2—TaqMan universal master mix II, GS7—GeneStudio 7 qPCR instrument, 7500F - 7500 Fast qPCR instrument. **Sheet 8: Detailed gene copy number results.** GCN—gene copy number, avr—average, SD—standard deviation, lwr—lower, upr—upper, CI—95% confidence interval, CV% —coefficient of variation %, DQ—dosage quotient, CGH—comparative genome

hybridization; wd—with dilution, wod—without dilution; g—good quality, p—population, b —bad quality; orange number or amb—ambiguous result, red number—misclassification, orange background—not all MLPA probes were used for a CYP21 gene due to chimeric gene, red background—MLPA was inconclusive probably due to double chimeric genes. **Sheet 9: Expected gene copy numbers.** GCN—gene copy number, avr—average, exp—expected, dev —deviation, mod—modified SD—standard deviation, lwr—lower, upr—upper, CI—95% confidence interval, CV%—coefficient of variation %, DQ—dosage quotient, CGH—comparative genome hybridization; g—good quality, p—population, b—bad quality; orange number— ambiguous result, red number—misclassification at GCN or a deviation above 0.4 at avr dev GCNs or a misclassification at rounded GCNs. **Sheet 10: Detailed results of linear discriminant analyses.** The estimated total integer gene copy number (GCN) differed from the input class in sample NA11839 (the only sample having inconsistency in its input classes) and in sample H005 which has a relatively low probability value of classification. Cross-validation, when the estimated integer GCN of one sample is estimated without its input class and is based on the data of all other samples, supported the estimation in the former case, and led to a different classification in the latter one. A deviated classification of *C4* genes was also observed by cross-validation in sample NA11839. The cross-validation of *CYP21* genes could not be performed in two samples (H004 and H010) with the rare 3 GCN of *CYP21A2*, because their classes consisted of only one sample. avr—average, P—probability, red number—inconsistent input class, orange background—low probability or difference between input, resultant or cross-validation class, yellow background—cross-validation cannot be applied, red background—ambiguous estimated integer GCN.
(XLSX)

## Acknowledgments

We are indebted to Mark Eyre for English proofreading. We thank Prof. Barna Vasarhelyi for ensuring an inspiring research environment. Otto Darvasi passed away before the submission of the final version of this manuscript. Marton Doleschall accepts responsibility for the integrity and validity of the data collected and analyzed.

## Author Contributions

**Conceptualization:** Márton Doleschall.

**Data curation:** Márton Doleschall.

**Formal analysis:** Márton Doleschall, Ottó Darvasi, Zoltán Herold.

**Funding acquisition:** Márton Doleschall, Attila Patócs.

**Investigation:** Márton Doleschall, Zoltán Doleschall.

**Methodology:** Márton Doleschall.

**Project administration:** Márton Doleschall, Gábor Nyirő.

**Supervision:** Márton Doleschall, Anikó Somogyi, Péter Igaz.

**Validation:** Márton Doleschall.

**Visualization:** Márton Doleschall.

**Writing – original draft:** Márton Doleschall.

**Writing – review & editing:** Márton Doleschall, Ottó Darvasi, Zoltán Doleschall, Gábor Nyirő, Anikó Somogyi, Péter Igaz, Attila Patócs.

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
