## [Decision Letter · Decision Letter 0]

17 May 2022

PONE-D-22-07409Quantitative PCR from human genomic DNA: the determination of gene copy numbers for CAH and RCCX CNVPLOS ONE

Dear Dr. Doleschall,

Thank you for submitting your manuscript to PLOS ONE. After careful consideration, we feel that it has merit but does not fully meet PLOS ONE’s publication criteria as it currently stands. Therefore, we invite you to submit a revised version of the manuscript that addresses the points raised during the review process.

We look forward to receiving your revised manuscript.

Kind regards,

H. Hakan Aydin, MD, FAACC

Academic Editor

PLOS ONE

Journal Requirements:

2. Thank you for including the following ethics statement on the submission details page:

'The current research was conducted with the approval by the National Scientific and

Ethical Committee, Medical Research Council of Hungary (TUKEB, ETT). Approval

number is 4457/2012/EKU.'

Please also include this information in the ethics statement in the Methods section of your manuscript.

3. Please provide additional details regarding participant consent. In the ethics statement in the Methods and online submission information, please ensure that you have specified (1) whether consent was informed and (2) what type you obtained (for instance, written or verbal, and if verbal, how it was documented and witnessed). If the need for consent was waived by the ethics committee, please include this information.

Reviewers' comments:

Reviewer's Responses to Questions

**Comments to the Author**

1. Is the manuscript technically sound, and do the data support the conclusions?

Reviewer #1: Yes

Reviewer #2: Yes

2. Has the statistical analysis been performed appropriately and rigorously? 

Reviewer #1: Yes

Reviewer #2: Yes

3. Have the authors made all data underlying the findings in their manuscript fully available?

Reviewer #1: Yes

Reviewer #2: Yes

4. Is the manuscript presented in an intelligible fashion and written in standard English?

Reviewer #1: No

Reviewer #2: Yes

5. Review Comments to the Author

Reviewer #1: This manuscript is presenting qPCR approached for determination of gene copy number variation for the CAH and RCCX CNV from human genomes. They use a lot a of great statistical analysis to evaluate the value, the standard deviation and abilities to many primers, assays set to evaluate it. Methods seem to be well done and will be interesting for the readers of the journal but more work need to be done to make this paper more clear for the reader. Many section need to be redone, they indicated in acknowledgments someone that help with the English but believe need to be done again. I will ID some not clear sections in my comments but more work need to be done to the entire manuscript. Not all acronym are defined well and this manuscript presents so many supplement materials important and cited in the text that is is also difficult to follow. More than 20 figures and 20 tables if we included the one not supplement in.

Specific comments:

Title: Please define CAH and RCCX CVN, CAH dont seem to be defined in the manuscript

Abstract: Define the RCCX CVN.

P3L46 to L53, rephrase this section not clear, you mention the DNA genomic template and GCN is a whole number, very not clear and difficult to follow.

P4L62-67 difficult to follow as well.

P5L104 need a dot after File). L 106 will change by Qiagen with "using a Qiagen ..."

P6L109, it was not clear here what it mean by "Purchase DNA samples ..." and believe those sample were the samples used for "Population" important part in the manuscript but difficult to understand, you purchase blood for your population studies? or DNA not sure I follow here?

All material and method is lacking on clarity.

L116 "DNA working solution" of what?

L117 not sure I follow what you mean by "in a separate parallel measurement? Clarify this paragraph?

L127 Do you need the double "M" in "FAMM"

L129 and over the manuscipt you need to add the brand or cie name, ex for the Quantstudio 7 qPCR system (QS7)

Define also RPPH1 reference gene. L134 Info on the Bioline qPCR reagents? What PCR profile did you use? Also not sure the S5-table is clear enough about the master mixes?

L159 Primer-BLAST need reference

L206 define the HERV-K(c4) CNV?

Was not easy to follow what exactly mean in your manuscript the "good quality" "population" "bad quality" and they are related to your Purchase DNA?

Figure S1: do you have a "r" or it should be a "r square"

S4-table: What is the information on dye and quencher used?

In abstract you mentioned it was tested with Southern, MLPA and aCGH was it discussed? Very short discussion.

Reviewer #2: The paper technical sound and do the data support the conclusions. Data analysis appropriately and rigorously and presented in an intelligible fashion and English language is need revision. The data underlying the findings described in their manuscript fully available without restriction, with rare exception . The data should be provided as part of the manuscript or its supporting information, or deposited to a public repository.

6. PLOS authors have the option to publish the peer review history of their article (what does this mean?). If published, this will include your full peer review and any attached files.

Reviewer #1: No

Reviewer #2: No

---

## [Author Response · Author response to Decision Letter 0]

30 Jun 2022

@Reviewer #1:

We thank the Reviewer #1 for valuable remarks. We think that the remarks have helped us to greatly improve the manuscript.

Please find our answers below.

@Many section need to be redone, they indicated in acknowledgments someone that help with the English but believe need to be done again.

The English has been revised by a native speaker biologist.

@I will ID some not clear sections in my comments but more work need to be done to the entire manuscript.

The manuscript has been revised, as proposed.

@Not all acronym are defined well

The absent definitions have been added.

@and this manuscript presents so many supplement materials important and cited in the text that is is also difficult to follow. More than 20 figures and 20 tables if we included the one not supplement in.

The first full version of the current manuscript was double as long as the submitted one. A lot of information was placed from the main text to the legend of supplemental materials because of the word limit required by many journals, but we have placed back the most important pieces of information. We have tried to balance the amount of provided information in the main text against its length.

@Specific comments:

The locations of changes are indicated according to „Revised Manuscript with Track Changes”.

@Title: Please define CAH and RCCX CVN, CAH dont seem to be defined in the manuscript

We have replaced “CAH” with “congenital adrenal hyperplasia” and “CNV” with “copy number variation” in the title. The exact definition of RCCX CNV is in S1 Table. We think the explanation for RCCX is too long and complicated to place into the title. CAH is defined in P8L140 of the manuscript.

@Abstract: Define the RCCX CVN.

We have added “a CNV” to the Abstract to help understanding. The length of the Abstract is limited, so the exact definition for RCCX is too long to add it on.

@P3L46 to L53, rephrase this section not clear, you mention the DNA genomic template and GCN is a whole number, very not clear and difficult to follow.

P4L62-67 difficult to follow as well.

We have rewritten the all section: “The qPCR for human GCN determination can be distinguished from qPCR for gene expression by some key features: 1.) Genomic DNA is the template. The template complexity, which can reduce the performance of qPCR, is much greater in genomic DNA than in total mRNA of a particular tissue: The haploid human genome consists of 3.1 billion base pairs, and millions of base pairs differ between two random haploid chromosome sets, while a few hundred genes account for 50% of transcripts in most human tissues covering only a couple of hundred thousand base pairs in total length.

2.) Limit of detection (LOD) is not crucial. The absolute copy number of a target gene in a DNA sample is proportional to the absolute number of haploid chromosome sets, which can be approximately calculated from the mass of genomic DNA in the sample. The absolute number of haploid chromosome sets can be more accurately determined by the quantitative measurement of a reference gene, which invariably occurs once in each haploid chromosome set. The ratio of absolute copy numbers of a target gene and a reference gene in the DNA sample of a subject is identical to the ratio of the copies of target and reference genes in two haploid chromosome sets of a diploid cell. The ratios of the target and reference genes is not conditional on the amount of genomic DNA in a sample, and the GCN of a target gene is easily calculated from this ratio since the GCN of a reference gene is always two in a diploid cell. Therefore, the amount of genomic DNA in a measurement also does not influence GCN (in theory), and can be chosen to be conveniently above the limit of detection (LOD).

3.) The differentiation between greater consecutive GCNs is difficult. The quantification cycle (Cq) is determined by qPCR to characterize the absolute copy number of a gene in reality. Cq is proportional to a relatively short DNA sequence specific to a target or a reference gene, and GCN is calculated from Cqs related to the target and reference genes. GCN determined by qPCR can be called “measured GCN”, and can be a positive real number (for example, a rational number), not necessarily a non-negative whole number. The relationship between Cq and GCN can be described by the equation: Cq(target gene)-Cq(reference gene) = -((log2(GCN)/log2(2))-1. The reference gene Cq is constant in theory, and therefore only the target gene Cq determine GCN. The theoretical difference between two target gene Cqs derived from two consecutive GCNs will approach zero if GCN approaches infinity. This means that the theoretical difference of two target gene Cqs is ∞ between 0 and 1 GCN, ΔCq=1 between 2 and 1 GCNs, ΔCq=0.585 between 3 and 2 GCNs, ΔCq=0.415 between 4 and 3 GCNs, ΔCq=0.322 between 5 and 4 GCNs, and so on. Therefore, it becomes more and more difficult to differentiate the greater consecutive GCN, which presents the key problem of qPCR as well as other molecular biology methods for GCN determination.

4.) The inaccurately measured GCNs can be easily identified in the majority of cases. Ambiguity is the state of a measured GCN which is not close enough to an integer GCN to assign unequivocally the measured GCN to the integer GCN. The measured GCN is a continuous variable, and therefore a measured GCN can be about halfway between two integer GCNs, which clearly indicates the inaccuracy of the particular measurement. The distribution of several measured GCNs derived from the same integer GCN approaches a normal distribution, resulting in the majority of the measured GCNs around the real integer GCN (unambiguous GCNs), some measured GCNs between the real GCN and an adjacent integer GCN (ambiguous GCNs) and a few measured GCNs around the adjacent integer GCNs (misclassified GCNs).”

@P5L104 need a dot after File).

We have added it to the text.

@L 106 will change by Qiagen with "using a Qiagen ..."

We have modified the text in P9L169 as it was requested.

@P6L109, it was not clear here what it mean by "Purchase DNA samples ..." and believe those sample were the samples used for "Population" important part in the manuscript but difficult to understand, you purchase blood for your population studies? or DNA not sure I follow here?

We have modified the text: “Genomic DNA samples were also purchased from the International Histocompatibility Working Group (IHWG).”

@L116 "DNA working solution" of what?

We have modified the text: “DNA working solutions with 5 ng/µl DNA concentration were separately diluted from the stock solutions of the DNA samples for 3 replicate measurements”

@L117 not sure I follow what you mean by "in a separate parallel measurement? Clarify this paragraph?

We have clarified the paragraph: “DNA working solutions with 5 ng/µl DNA concentration were separately diluted from the stock solutions of the DNA samples for 3 replicate measurements, except for ones for positive controls (more than 3 separately diluted working solutions) and for the calibration curve (a series of dilutions). DNA samples derived from our own subjects were divided based on DNA quality into “good quality” (A260/A280>1.8 and A260/A230>2.0 and no sign of DNA degradation) and “bad quality” study groups (n=10). The SD039 reference sample for MLPA was assigned to the “good quality” group (n=17). The DNA samples purchased from IHWG were labeled as the “population” group (n=19).”

@L127 Do you need the double "M" in "FAMM"

FAM usually stands for fluorescein amidites in molecular biology, which are important synthetic equivalents of fluorescein dye, and used for labeling oligonucleotide probes. We think it would be ideal to reserve this acronym for the dyes.

@L129 and over the manuscipt you need to add the brand or cie name, ex for the Quantstudio 7 qPCR system (QS7)

We have removed the information from S3 Table and added it to the text as it was requested.

@Define also RPPH1 reference gene.

We have defined it in P10L200.

@L134 Info on the Bioline qPCR reagents?

We have added it to the text in P11L220.

@What PCR profile did you use?

The qPCR profiles were used according to the manuals of manufacturers. We have added the information to the text in P11L205-6.

@Also not sure the S5-table is clear enough about the master mixes?

We think all of the information is provided now in Materials and Methods section (P10L195-P11L205), S5 Table and the legend of S5 Table.

@L159 Primer-BLAST need reference

We have added it to the text in P12L232.

@L206 define the HERV-K(c4) CNV?

We have added it to the text in P6L116, and we have also added some additional information to the S1 Table.

@Was not easy to follow what exactly mean in your manuscript the "good quality" "population" "bad quality" and they are related to your Purchase DNA?

They are study groups as described in P10L189-193. The study groups enable us to yield more detailed insights. For example, the estimated accuracies of study groups significantly differed only in certain qPCR assays as shown in Fig. 4. The qPCR for GCN determination in general has been claimed by an article (PMID: 21364933) to be very sensitive to quality of the genomic DNA, generating systematic biases. However, this conclusion has been drawn from one qPCR assay. Our results confirmed this conclusion only partly, and also pointed to that the experimental results based on one assay should not be generalized.

@Figure S1: do you have a "r" or it should be a "r square"

The “r” stands for Pearson’s rank correlation. We have clarified it in the legend of S1Fig. The “r” is directly related to the coefficient of determination “r square” in the obvious way. “r” is a number between -1 and 1 which characterize the direction of the correlation (negative or positive), whereas “r square” is a number between 0 and 1 without the directional component.

@S4-table: What is the information on dye and quencher used?

The custom Taqman probes, used in the current study, contained a 3’-nonfluorescent quencher and a 3’-minor groove binder. We have not found the exact chemical compositions for them. We have added the information to the text in P10L198-199.

@In abstract you mentioned it was tested with Southern, MLPA and aCGH was it discussed? 

We have added the next sentences to the text: “Furthermore, all these unambiguously estimated integer GCNs were in 100% concordance with the integer GCN estimations from MLPA and the findings of Southern blot and array CGH from previous studies.” and ”Southern blot was the first GCN determination method for RCCX CNV [43], and uses a multiplex approach because the unlabeled genomic DNA fragment pattern bound to the membrane can be examined with several probes for the elements of RCCX CNV in succession. The disadvantages of Southern blot include high labor intensity, high time demand, and only semi-quantitative GCN results based on a human operator’s evaluation [44], decreasing its suitability for the genetic test of CAH. Array CGH is a high-throughput method, but its high labor intensity, high time demand and high cost do not fit to the needs of CAH laboratories, where the vast majority of array CGH results would not be used.”

@Very short discussion.

We have largely extended the Discussion.

@Reviewer #2

We would like to thank the Reviewer for the remarks.

@English language is need revision.

The English has been revised by a native speaker biologist.

@The data underlying the findings described in their manuscript fully available without restriction, with rare exception . The data should be provided as part of the manuscript or its supporting information, or deposited to a public repository.

We have added the raw MLPA data and the robustness experiments data of qPCR to the minimal data set in S1 File. Our minimal data set will be also available in Zenodo database (DOI: 10.5281/zenodo.6780358) after the publication of the current paper.

---

## [Editor Report · Decision Letter 1]

25 Oct 2022

Quantitative PCR from human genomic DNA: the determination of gene copy numbers for congenital adrenal hyperplasia and RCCX copy number variation

PONE-D-22-07409R1

Dear Dr. Doleschall,

We’re pleased to inform you that your manuscript has been judged scientifically suitable for publication and will be formally accepted for publication once it meets all outstanding technical requirements.

Kind regards,

H. Hakan Aydin, MD, FAACC

Academic Editor

PLOS ONE
---

## [Editor Report · Acceptance letter]

8 Nov 2022

PONE-D-22-07409R1 

Quantitative PCR from human genomic DNA: the determination of gene copy numbers for congenital adrenal hyperplasia and RCCX copy number variation 

Dear Dr. Doleschall:

I'm pleased to inform you that your manuscript has been deemed suitable for publication in PLOS ONE. Congratulations! Your manuscript is now with our production department. 

Kind regards, 

on behalf of

Professor H. Hakan Aydin 

Academic Editor

PLOS ONE